# Danish PV Prosumers' Time-Shifting of Energy-Consuming Everyday Practices

**Kirsten Gram-Hanssen** *, **Anders Rhiger Hansen** and **Mette Mechlenborg**

Department of the Built Environment, Aalborg University Copenhagen, 2450 Copenhagen SV, Denmark; arhansen@build.aau.dk (A.R.H.); mme@build.aau.dk (M.M.)
* Correspondence: kgh@build.aau.dk; Tel.: +0045-2360-5653

**Abstract:** Consumer engagement in the energy system is necessary to ensure a low-carbon transition. However, this has proved difficult because consumers are engaged in pursuing everyday practices rather than focusing on abstract questions of energy. Recent studies have suggested that being a prosumer can make a difference. This paper builds on survey data from a representative sample of 2505 photovoltaic (PV) owners in Denmark combined with 12 qualitative in-depth interviews. The results indicate that PV owners consider that they have become more concerned about energy consumption and adjust the timing of their everyday practices to their production. Thus, 67% of the households 'often' or 'always' time-shift the use of washing machines to their production. The extent to which households time-shift is strongly related to their net-metering scheme. Thus, 75% of the households on hourly metering stated that they 'to some' or to 'a great extent' adjust their consumption, compared to only 26% of the households on annual metering. This financial effect is interpreted in an everyday life context where financial gain transfers meanings of self-sufficiency and sustainability, rather than primarily being viewed as rational economic behaviour. The conclusion discusses the policy implications of methods to engage the consumer.

**Keywords:** residential electricity consumption; energy prosumers; renewable energy production; load management; demand-side management; household practices

## 1. Introduction

It is increasingly acknowledged that the active participation of consumers is a premise when creating a sustainable energy system with a high proportion of renewable energy. This active participation can be in the form of demand-side management (DSM) in which households are engaged in reducing and time-shifting their consumption, or it can be in the form of consumers producing energy (e.g., through a solar photovoltaic (PV) system) and thus becoming prosumers [1]. Both cases involve the active engagement of consumers, and both can be considered part of a smart energy system, where production and consumption are digitalized so that different actors in the grid can take part in handling the energy system, including securing matching consumption profiles for production. Consumers are acknowledged as important, although a lack of products and service designs that can support consumers in their new roles has also been noted [2]. Recently, research has been conducted in developing methods and tools to enhance DSM and prosumption in the smart grid [3–6].

The smart grid, including its tools and methods, is in the process of being developed, and household practices are changing along with new products and methods. In developing the future smart grid, more knowledge of how households react in their everyday lives is relevant, specifically knowledge about the extent to which households time-shift their energy consumption to match their production.

When consumers shift their electricity consumption in time, they make changes in their everyday routines and practices. In recent years, considerable research has been published in the field of

energy consumption and everyday practice, especially within a practice theoretical research tradition (see, e.g., [7–12]). Within theories of practice, the individual consumer does not rationally and consciously decide on his or her actions but participates in practices that are collectively shared, routinized, and based on norms of how to live a normal life. In addition, energy is consumed as part of the performance of everyday practices, such as cooking or laundering. Energy is thus not something that the consumer has a direct interest in but is something that happens because they perform other activities, which are important to them. This has a major effect on understanding how smart grid technologies and time-shifting consumption are integrated into the everyday lives of households and on understanding the importance of having PVs concerning the household's relation to energy consumption. Specifically, it has been demonstrated that laundry and dishwashing, in particular, can be time-shifted [13] but that time-shifting these practices is largely related to the family's other routines [14,15] and that many practices in everyday life follow both personal and societal rhythms [16].

In terms of the extent to which households engage in time-shifting, the PV prosumer is a relevant case because previous studies have suggested that PVs are among the microgeneration technologies with the greatest potential for giving householders a sense of being involved in energy production [1]. Several studies have addressed the extent to which PV owners are time-shifting their consumption to their production. A case-based study from the UK concluded that this largely depends on the size of the household consumption and whether residents are home during the daytime and that it also matters how PVs are introduced to the household by installers and other professionals [17]. Another UK-based study revealed that consumers behave differently according to why they bought the PV system and that consumers who have invested in PVs primarily based on economic reasons also take an economic approach to buying and selling electricity [18].

A Swedish study concluded that prosumers time-shift their consumption to a limited extent to match their production; however, the study also reported that little or no economic incentive existed to do so among the surveyed prosumers [19]. The same study documented that prosumers become more interested in energy issues, even many of them already before acquiring the PVs also was engaged. Households, for instance, talk about being 'nerdy' and enjoy following how much electricity they are producing [19]. A qualitative study from Denmark highlighted a strong interest in consuming ones' own electricity and related this to an accounting scheme that made the prosumers feel they were giving their electricity away for free if they were not using it themselves [20]. Furthermore, this study found that some households started to use more of their 'free' electricity rather than 'giving it away', as they expressed it [20]. A study from Wallonia showed that as many as 40% of a surveyed sample of PV owners claimed to time-shift to their production notwithstanding the lack of financial incentives [21]. A German study concluded that the accounting schema and environmental motivation can explain the differences in the extent to which households time-shift [22].

Thus, the literature suggests that a tariff structure influences time-shifting among prosumers; however, it also suggests that tariffs only present part of the understanding. Bringing this together with understanding from theories of practice, the meanings associated with time-shifting mundane everyday practices must also be further developed.

Studying PV owners and their time-shifting practices should preferably include data that vary according to the tariff structure. In Denmark, PVs were established in different periods, and following this, they also run with different metering schemes [23]. This makes it possible to include households with PVs with very different financial incentives in the same survey. The first PV in Denmark was established as an experiment in 1993, and until 2010, the majority of all installed PVs were part of different types of demonstration projects with a subsidy. These households had an annual net-metering scheme, meaning that the households could consume from the grid and produce to the grid, and be billed once a year on the difference between the two, which means that the households had no financial gain from self-consumption. The yearly net-metering scheme became permanent in 2010, although at a lower price for selling compared to buying. The vast majority of the PV owners in Denmark bought their PVs between 2010 and 2012, at a time when the economic conditions for establishing PVs in

Denmark were favourable due to the lower market prices for the PV system and the attractive annual metering scheme.

In 2012, an emergency intervention from the government put an end to these conditions for PV owners, as it resulted in massive economic loss for the state due to large reductions in the tax revenue. This was followed by an almost immediate stop of new PV installations, and it has only been slowly growing since then. All PVs installed after 2012 have been on an hourly or immediate net-metering scheme, where the household surplus of electricity of up to 6 kW from solar panels is sold at a fixed price of 0.6 DKK/kWh, whereas the price of the electricity they buy is approximately 2.20 DKK/kWh. Whether households have hourly or immediate net metering depends on the type of meter and the local utility, although the economic difference is not great [23]. The hourly and immediate net-metering schemes are now also starting to apply to households who originally had annual accounting; thus, these households must soon accept changing their metering scheme.

Denmark represents a relevant case for studying time-shifting for self-consumption among PV owners because different groups of PV owners can be included according to their metering scheme. Concerning PVs in Denmark, in an international context, Denmark is a sustainability front-runner regarding energy efficiency and is a major consumer (e.g., among countries with the highest carbon dioxide emissions per capita) [24]. Thus, from an international perspective, the results from Denmark may be comparable to other north European countries with similar policies [25], whereas the situation in the global south may be very different [26]. Cross-cultural studies in highly different socio-demographic countries can reveal how both the social organization of everyday life and the socio-material setup of the energy infrastructure influence questions of time-shifting to a large extent [15]. However, the present study only includes one specific country, and the results should be interpreted accordingly.

This paper aims to investigate the extent to which Danish PV prosumers time-shift their everyday practices and why they do it, including the importance of the metering scheme and what the financial incentive means to them. The paper first reports on the methods used in the project, followed by the results and discussion.

## 2. Materials and Methods

In this paper, the data are based on a mixed-method approach using a survey questionnaire and qualitative in-depth interviews with households using PVs. The advantage of this approach is both to have objective quantitative input from a representative sample of all PV owners and to gain a more detailed and context-based explorative understanding of the motivations and practices among PV owners. Parts of the presented material are also described in two Danish reports on the survey questionnaire [27] and qualitative material [28]. Further, another paper has been published on the survey results, highlighting the different ways of being prosumer [29]. This paper presents new results because it provides a new survey analysis and combines quantitative and qualitative methods in answering the specific question regarding the extent and reasons prosumers time-shift energy consumption.

The survey questionnaire includes a representative web survey of Danish households that have installed PVs. The survey was conducted by Statistics Denmark on behalf of Aalborg University during the fall of 2018. The addresses of all Danish PV owners were known from the registration data. This provided a total population of 72,900, and from this population, a representative sample of 4567 households was selected and contacted. To secure a balanced share of households on different net-metering schemes, the hourly and immediate net-metering schemes were over-sampled. A response rate of 55% resulted in 2505 responses, but due to the missing data on some questions, the number of observations is slightly lower in some parts of the analysis. The average effect of PV systems was 4.7 kWp, with a standard deviation of 1.4. The survey data were combined with data on the net-metering scheme from Energinet.dk, which registers information on Danish PV owners. The questionnaire focused on the everyday life of people living in households with PVs, including questions on the

acquisition of the PVs, engagement with energy issues, time of use, and time-shifting. All analyses of survey data were conducted using the statistical program Stata.

Qualitative data include 12 interviews with households having PVs. Contact with households was provided through two net companies participating in the research project. Six households were in Zealand, and six households were in Jutland. Contact procedures were initially designed to select a variety of households according to socio-economics and the acquisition time of PVs. New rules from GDPR (EU general data protection regulation), however, restricted the ways companies could contact their customers and provide contact information, meaning that other means of making contact had to be initiated. In Jutland, the net company used more personal contacts to households, which had been part of previous projects or were already known in other ways by the company as being interested in being interviewed. These interviews are slightly biased by people having had PVs for longer periods and being more reflective about having them. In Zealand, the contact procedure included the use of social media through which households contacted the net company to state their interest in participating. Selection bias includes the self-selection of those interested.

The demographic variation among the informants was sought when selecting households for interviews, even with the selection being restricted by the contact procedure. Four of the interviews were with couples. One interview was with a couple living apart, but both partners had PVs. Thus, the 12 interviews comprised 18 informants and 13 PVs. Of these informants, six were women and 12 were men. Moreover, 12 out of the 18 were pensioners, and three households had children living at home. For more details, see Table 1.

**Table 1.** Overview of the socio-demographics of the 12 qualitative interviewees. A *p* after the age indicates that the person is a pensioner.

| Interview | Pseudonym | Age (m) | Age (f) | Background (m) | Background (f) | Children |
|-----------|-----------|---------|---------|----------------|----------------|----------|
| A | Anders & Anja | 67p | 69p | Machine operator | Secretary | |
| B | Børge & Bente | 73p | 69p | Electrician | Housewife | |
| C | Carl | 70p | | Phys-teacher | | |
| D | Dennis & Dorit | 70p | 71p | Engineer | Secretary | |
| E | Erik | 53 | | IT consultant | | X |
| F | Frank | 53 | | Shop owner/porter | | X |
| G | Grethe | | 57 | | IT consultant | |
| H | Hanne & Helge | 72p | 75p | Phys/math teacher | Local politician | |
| I | Ivar | 62 | | Building constructor | | |
| J | Jens & Jette | 66p | 67p | Janitor | Seamstress | |
| K | Knud | 68p | | Building constructor | | |
| L | Lars (& Lærke) | 47 | 42 | Surveyor | Unknown | X |

Notes: IT: information technology; Phys: physics; m: male; f: female.

The qualitative interviews should not be regarded as representative, although the socio-demographics of the interviewed households do not differ substantially from the whole population of Danish PV owners [23], except that many have had their PVs for a longer time, are a bit older than average, and most have the annual metering scheme. Thus, 10 of the PV owners were on the annual metering scheme. One participant did not know the scheme. One was on a company contract, and one was on hourly metering. However, most of them had been told that they must soon change the metering scheme according to the new rules. Interviews were conducted as open semi-structured interviews [30]. The interviews lasted between 1 and 2 h, and they were recorded, transcribed, and analysed. The analysis included questions about the motivation for acquisition, understanding of and relation to the energy system, and everyday practices concerning electricity consumption.

## 3. Results

The results presented in this paper focus on the question regarding whether PV prosumers time-shift their everyday practices, and if so, why they do it and what it means to them. The results are divided into three subsections to answer the following questions: (1) Do prosumers time-shift their

electricity consumption, and if so, what is changed? (2) Do prosumers time-shift their consumption, why or why not? (3) Do households change engagement related to energy by being prosumers?

### 3.1. Do Prosumers Time-Shift Their Electricity Consumption and If So, What Changed?

In the questionnaire, we asked whether prosumers adjusted their use of different appliances according to their production, and the results are presented in Figure 1. For dishwashers, washing machines, and tumble dryers, more than half of the respondents stated that they often or always adjusted their use. Even for vacuum cleaning, more than a third stated that they often or always adjust the timing to the production of their PVs. For cooking and charging appliances, however, this seldom happens.

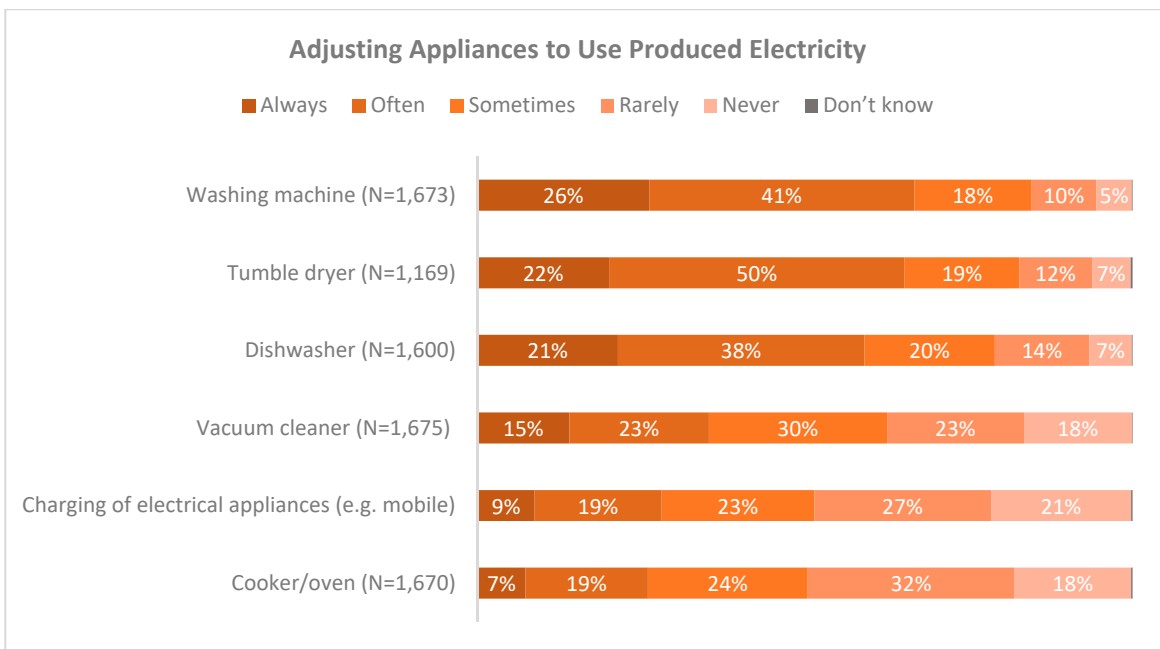

**Figure 1.** Responses to the question 'How often do you adjust your use of the following appliances to utilize your own produced electricity (for example using a timer)?' *n* = 2477. Percentages below 5% are not reported. The full table is in Appendix A.

Turning to the analysis of the qualitative interviews, two different approaches were distinguished among PV owners regarding how they relate to energy in general and to consuming their "own" electricity. These are the minimum adaptation and maximizing strategies [28].

Regarding minimum adaptation, for the interviewees, the PVs had resulted in minimum practice changes in everyday life, which included easy changes, adaptable solutions that did not require great technical insight, investment, or a radical change in daily life. The threshold for this minimum adaptation is set by different conditions (e.g., the net-metering scheme, routines, and habits of the household regarding work, children, and other interests, and understanding related to what is important for a good life). These households stated that they think they should behave differently and adapt more than they do (by saving energy or time-shifting). Hans stated, 'Maybe I should, but it doesn't really bother me much'. Similarly, Ivar spoke of 'comfort', 'laziness', and 'ugly questions' when we asked questions concerning changing practices because of being a prosumer. Along the same line, households representing a minimum adaptation turned to demonstrative images to justify their choices.

> 'So, I also don't want to say to the children that now you must not use your computer because it uses electricity, right?' (Lars).

Among households conforming to this minimum adaptation strategy, all of the households perform activities related to time-shifting or reducing energy consumption. However, this is often a subject of negotiation, and the interviews included both excuses and rejections, as seen in the quotes above.

In contrast, maximization strategies for using one's own PV production require additional resources, which could be practical, technical, or economic. The everyday life of the household is subject to changes and negotiations. Two variants include a low-technology approach, which is based on everyday inventions and realignment of practice, and a smart technology approach, in which technology is used to take further steps towards energy optimization and using one's own produced electricity.

Research has shown that, when consumers move their electricity consumption over time, it is about making changes in their everyday routines and practices [14]. However, interviews in this project indicate that several informants have sought to streamline their electricity consumption through software and flexible technological solutions, which, in their view, do not require everyday life to be changed. One interviewee, Erik, explained that he views different types of technology as a way to solve the resource problem. Erik is an IT consultant and has a Tesla and a German battery that can store far more power than is allowed in Denmark. He has suggestions for different timers that can work with flexible consumption in the home and for hydrogen tanks that can store energy. He explained that he is only in the starting phase and is open to the possibilities when they arrive.

In addition, Grethe is in favour of technical solutions to replace routines. Like Erik, she works professionally with technology but without a long education. She has installed various technologies in the home, not only related to using produced electricity but also related to other types of energy use. For her, it is about comfort and being able to routinize her practices around saving actions, so that they do not have to be conscious of their usage all the time. For example, she says she has a meter installed in the wood-burning stove, which beeps when it needs more firewood. Therefore, she does not use too much firewood or need to start over, due to the fire going out. In this way, she views it as a technique to increase the efficiency of firewood consumption:

> If I can save nature for … if I have to use three pieces of firewood, and I only use one instead of three, then I think it's nice. It is … I would not say comfortable, but it is also nice that I do not watch all the time … (Grethe).

Lars explained how he prepared his household for the new hourly net-metering scheme by investing in timers for the washing machine, dryer, and dishwasher. He also invested in smart thermostats on the radiators, although these are connected to district heating and thus are not related to electricity production. For Lars, it is very much about comfort, with technology as a co-producing factor so that he does not need to dedicate his attention and time to managing the consumption. These informants have several suggestions for technology that can make their consumption more flexible concerning PV production, including timers on home appliances, new ways to store power, and software that can control electricity consumption at different times of the day. This points to a specific informant profile as a kind of first mover with a strong technology interest.

The other version of the maximization strategy, the low-technology practical approach, can be exemplified through Frank but is also represented by other interviewees to a lesser extent. Frank is willing to put forth effort to save electricity and calls himself a 'nerd' (i.e., he goes into minute detail). He uses electricity only when the sun is shining. When it is cloudy or winter, he always considers alternatives, such as watching television at the fitness centre instead of at home, or he moves activities to another day. This extreme self-discipline and inventor personality in saving energy is about economics and a sense of freedom.

The notion of being able to minimize and manage one's resources is an important aspect of the prosumer identity. However, Frank also relates to the larger question of what he leaves to the next generations, exemplified by his son. Underneath his saving ambitions is a great fear for the future (his own and the world's), which is demonstrated in anecdotes about divorce and bankruptcy and news of the climate crisis. The actions he takes, according to his calculations, results in producing 61.9%

of his own electricity consumption, of which he is very proud. He measures the effect of each action and can accurately reference what works best. Frank has also started sleeping in a caravan during the summer. Thus, he does not have to turn on a light or tempt himself to turn on appliances. He views it as a small vacation. Many of his practices use low technology and are practical, as exemplified when he discussed the refrigerator:

> Then, I have nerded … Yes, I use the word a little … So I have thought, 'how can you do [it] if you cannot make the energy yourself?' Then, I have my fridge filled with a lot of soda, and then, it runs high while the sun is shining, and then I have a thermometer and such; then I just turn it off at night. [ … ] Then, it runs clockwise. [ … ] If there are many kilos inside such a fridge, it will keep the temperature. And if there is something that needs to be defrosted, it also just goes in there at night (Frank).

Frank's maximization strategies are based on many different, low-cost, everyday actions that do not require new technology or major investments. Rather, this is a continuous discipline of tasks and habits, controlled by the energy calculator and great ingenuity, where Frank include discipline and great creativity in developing new ideas. However, Frank lives alone except for a son who is with him half the time.

In both cases of the maximization strategy (low or high technology), it is interesting to see that not only electricity is in focus, but the household resource consumption in general are included. Furthermore, the strategies are often driven primarily by one member of the household. These different methods to engage in consuming one's own produced electricity should be considered examples of strategies, which often overlap in reality, and in some cases, the same households may be viewed as drawing on different parts of these strategies in different situations.

*3.2. Do Prosumers Time-Shift Their Consumption, Why or Why Not?*

The question regarding the reasons for whether households time-shift relates to the question of minimum or maximum strategies for adjusting to one's production. Some households think they should do it more than they currently do but do not time-shift because of many practical issues in everyday life. In contrast, they may time-shift because there is satisfaction in mastering this issue. The question regarding the reasons for whether households time-shift was asked more directly in the survey. Figures 2 and 3 illustrate the results.

In Figure 2, the reasons people indicate that they time-shift their practices are almost equally split between economic, environmental, and self-sufficiency aspects. Being home during the daytime and fitting into the family routines are also key. Similarly, Figure 3 reveals the reasons for not time-shifting. Nobody being home during the daytime and not fitting well with the daily routines are the most prevalent answers for not time-shifting. In addition, the issue of economic gains being too small is observed in Figure 3 to have (high or some) significance for approximately one-third of the respondents.

Another way to analyse the issue of how much the economic gain means in the question of time-shifting consumption to one's production can be seen when comparing what time during the day households state that they use their appliances and relating this to their accounting scheme. Figures 4 and 5 illustrate a clear difference in the use of both washing machines and dishwashers, where households on hourly and immediate net-metering use their appliances to a higher degree during the daytime, whereas households on annual net metering, especially regarding dishwashers use appliances during the evening to a higher degree. Households on hourly or immediate net metering have an economic interest in using electricity when their PV system produces it, in contrast to households on an annual scheme. This economic interest demonstrates a clear correlation to the time of use of these two types of appliances, especially the use of dishwashers.

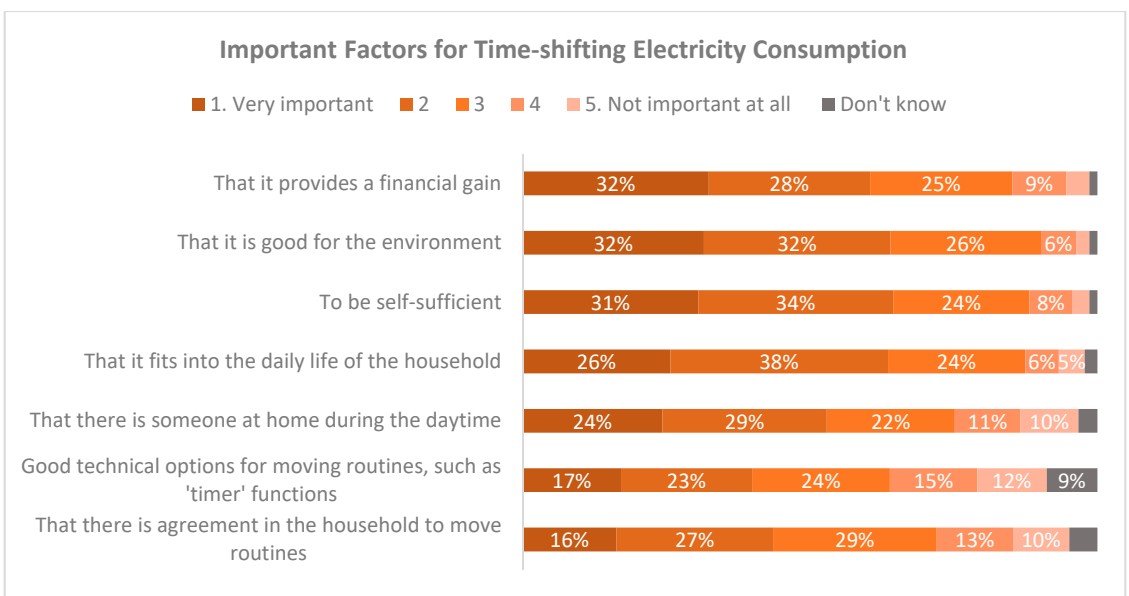

**Figure 2.** Responses to the question 'How important is the following for your household to move the time of using appliances to utilize your own produced power?' Questionnaire asked to indicate a number from 1 (very important) to 5 (not important at all). *n* = 1674. Percentages below 5% are not reported. The full table is in Appendix A.

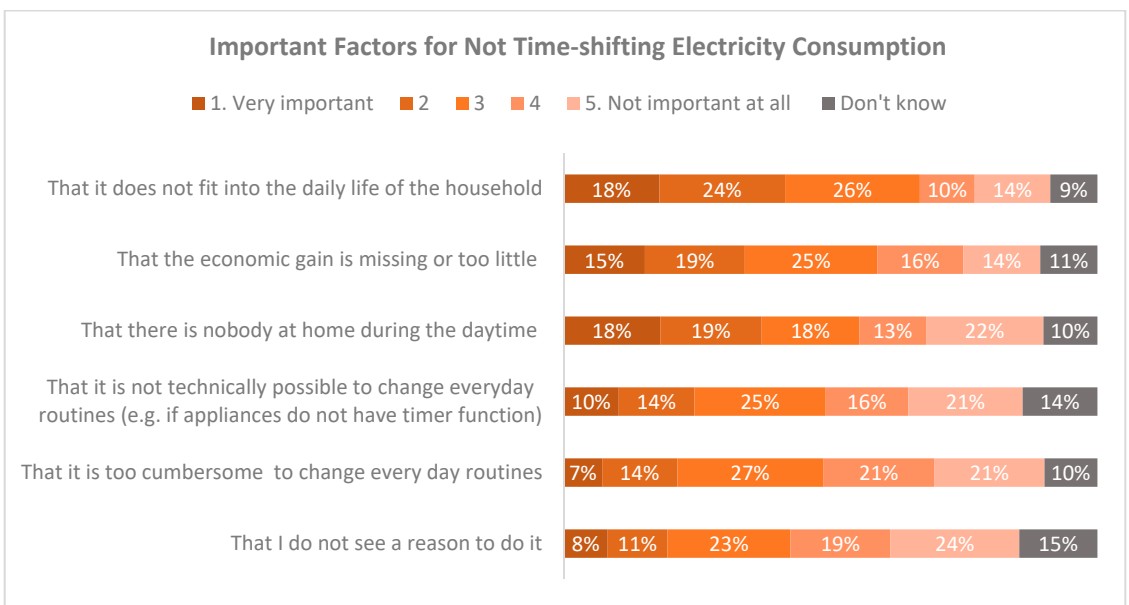

**Figure 3.** Responses to the question 'How important is the following for your household not to move the time of using appliances to utilize your own produced power?' Questionnaire asked to indicate a number from 1 (very important) to 5 (not important at all). *n* = 2834.

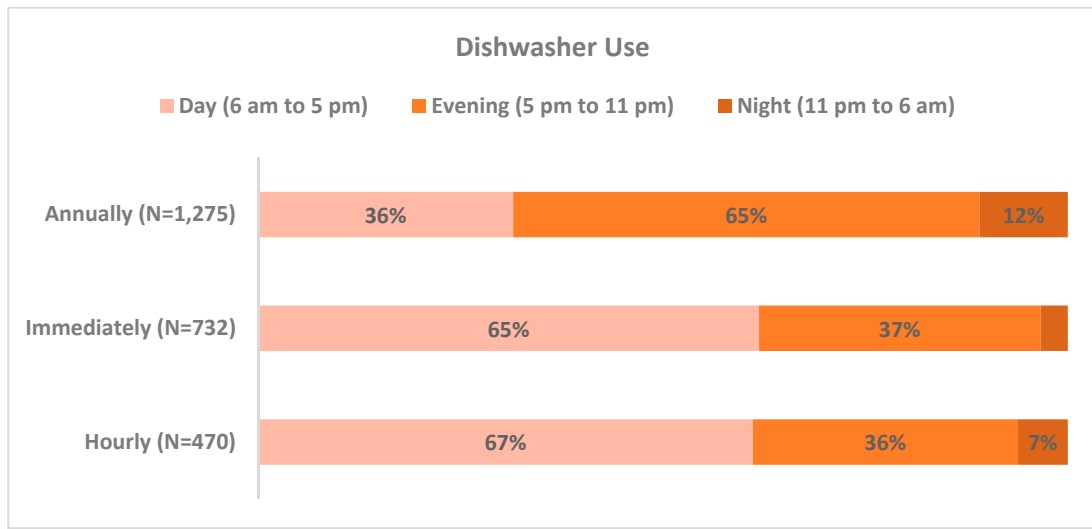

**Figure 4.** Responses to the question regarding when households report using a dishwasher across the net-metering scheme. Percentages below 5% are not reported.

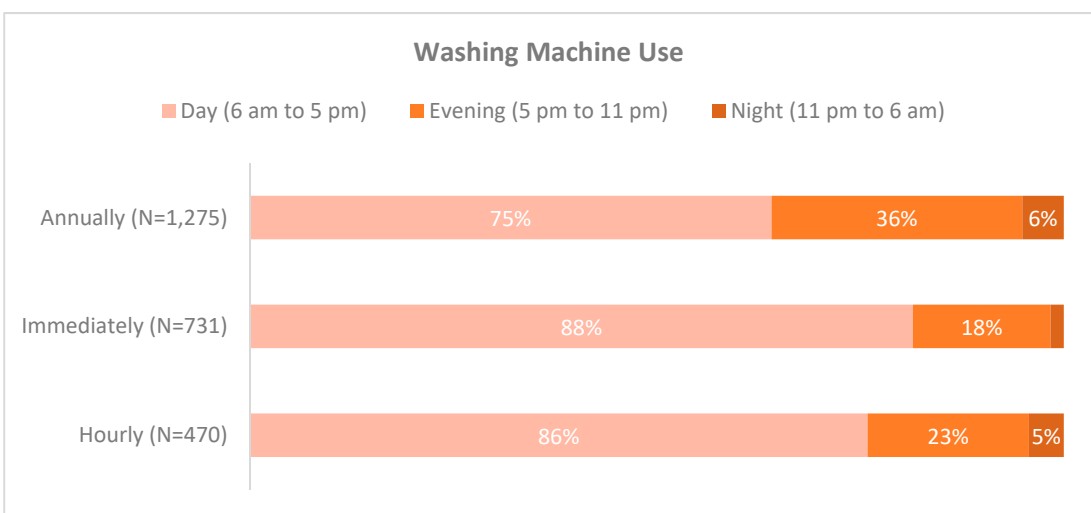

**Figure 5.** Responses to the question regarding when households report using a washing machine across the net-metering scheme. Percentages below 5% are not reported.

Answers to the question regarding when households use their appliances compared to their accounting scheme can be analysed in Figures 6 and 7. The questions regarding whether the households adjust their consumption to either their own production or to peaks in the electricity grid are compared in terms of their metering scheme (taken from the registration data). A significant correlation is found indicating that households on annual metering adjust their consumption to their production to a much lesser degree; however, some make adjustments even though they receive no economic gain. Furthermore, Figure 7 indicates that no clear correlation exists between the net-metering scheme and the extent to which the household adjusts their consumption to peaks in the electricity grid. Moreover, Figure 7 reveals that only 17% of all PV owners adjusted (to some or a large extent) their consumption to peaks in the grid. This is surprising, given that half of the respondents stated that they 'have gained a stronger interest in the Danish energy system' (Figure 8). One could then expect that a larger number of households would adjust their consumption to nighttime (for example) to help the energy system. Only a minute number of respondents answered 'don't know', indicating that the vast majority of households understood the question and have an understanding of what it requires to adjust their electricity consumption to peak demands in the electricity grid.

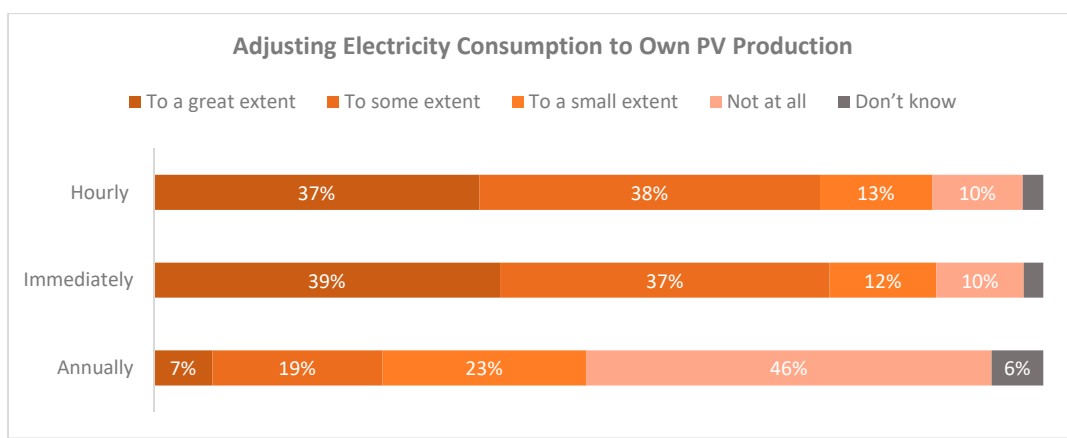

**Figure 6.** Responses to the question 'To what extent does your household adjust your electricity consumption to your own PV production' (*n* = 2477) related to the metering scheme. Percentages below 5% are not reported. The full table is in Appendix A.

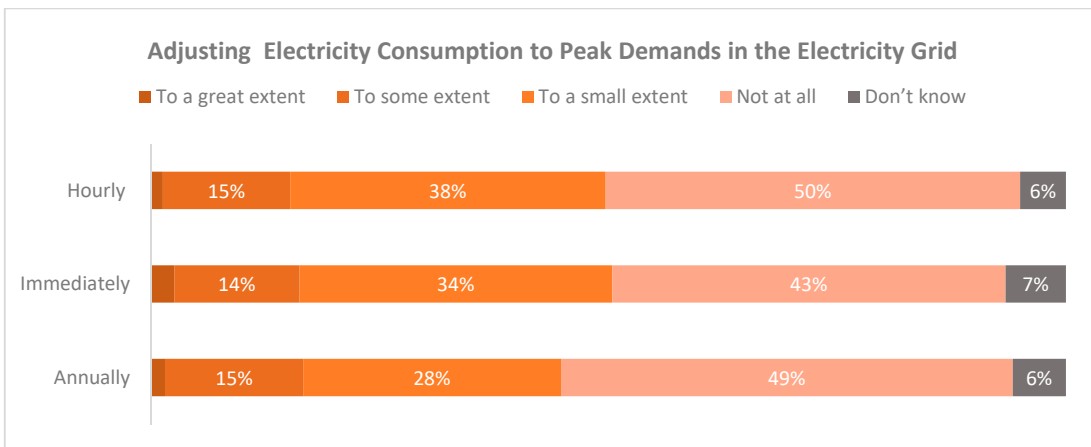

**Figure 7.** Responses to the question 'To what extent does your household adjust your electricity consumption to peak demands in the electricity grid' (*n* = 2477). Percentages below 5% are not reported. The full table is in Appendix A.

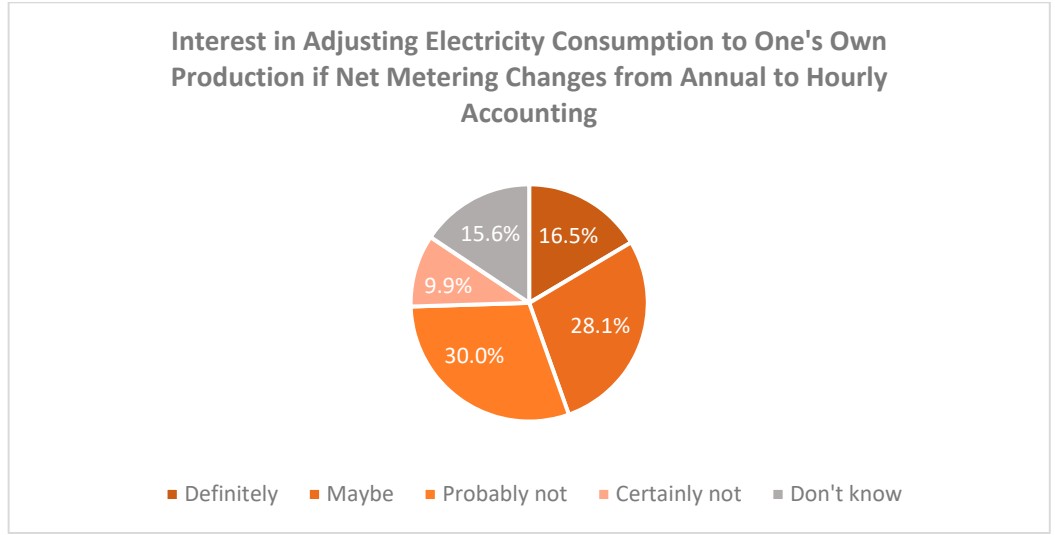

**Figure 8.** Answers from survey respondents who are now on yearly accounting to the question 'If you have an hourly or immediate net metering scheme (instead of a yearly scheme), do you think it would make you adjust your electricity more toward PV production?' *n* = 1275.

In the qualitative interviews, the question of the metering scheme was mentioned by all interviewees as a crucial parameter. Virtually all informants with annual metering did not see any reason to change their time of consumption. Several explained in different ways that 'it made no difference'. In most cases, the argument was whether the extra engagement would pay off financially on the electricity bill or in larger green accounts. However, households with annual net metering faced transition or were recently transitioned to hourly net metering. They were uncertain whether they would see significant changes in their electricity bill. In addition to the general frustration and disappointment with the policy in the field (Interviewees A, B, D, E, F, and I), the new net-metering scheme was a 'trigger' to reassess their approach to electricity generation and consumption. Several informants mentioned the shift as a new context for their everyday life with the PV system. Grethe stated, 'I am constantly aware of my energy consumption. I am. And especially here when the net-metering scheme has ceased. It annoys me boundlessly'.

Several indicated that, for this reason alone, they had taken the initiative to move their consumption so that they could better use their PV production (Interviewees G, L, and A). For example, when asked if the household had done anything about their method of using electricity, Dorit stated,

> Only this year'. [ . . . ] That's because . . . now we have to pay the money to [name of utility] first, and then we don't know if we will get anything back. We do not know that until the end of the year. We expect to get our money back, but we don't know (Dorit).

Although the utility makes information available via an application, the interviewees expressed that the information is not sufficiently adapted to be used, perhaps because it is not linked to the household's everyday life. The uncertainty has at least meant that Dorit has begun to organize tasks concerning the weather, especially for the dishwasher and washing machine. If the sun is not shining, then she has started to check the inverter to see what the PV is producing. If it produces below average, she postpones the task. She is willing to postpone a wash for several days, solely because she and her husband no longer know their energy accounts.

> . . . Because of the uncertainty of settlement . . . If the production is low, then I will probably look at the weather forecast and see if it got better the next day . . . It does not matter . . . [ . . . ] sometimes I think, it is not necessary to wash the clothes right now (Dorit).

Many of the informants expressed uncertainty about what the new metering scheme will mean for them economically. Thus, adaptation to another metering scheme cannot merely be considered a rational calculation of financial gain but should also be considered concerning risk management in household budgeting. Even informants in the qualitative interviews were concerned about this shift from yearly to hourly accounting, the results from the survey on this question indicate that not all households are equally concerned. As shown in Figure 8, only 16% of households are certain that the changing accounting scheme will make them adjust more towards their production.

Both qualitative and quantitative data provide a clear view of the importance of the net-metering scheme and its incentive to time-shifting consumption according to self-production. To understand the effect of the prices, it may be relevant to know approximately how much can be saved by time-shifting. As an example, the electricity for washing one load of laundry is about 2 DKK (0.27 EUR). Thus, the actual savings from time-shifting each load is less than 1.6 DKK. With an estimated 220 loads of laundry a year, this amounts to approximately 350 DKK (48 EUR), although only if all washing load are time-shifted. Whether this is considered much or little, depends on many things and varies among families.

A general point in all interviews is that finances cannot be separated from other motivations, even though finance plays a significant role related to buying PVs and time-shifting practices. Rather, financial gain (large or small) is experienced as a kind of reasoning for another purpose not directly concerning finances. That is, financial arguments underpin the aim, for example, to become self-sufficient or to reduce emissions related to climate change. For most of the interviewees, the motivation is far more

complex when delving further than the immediate financial gain. In addition, none of those interviewed made any clear calculations regarding what they could save by changing practices, and, for some, it was as much about the uncertainty in the economy as it was about the actual gain or loss. One interviewee made some calculations (Frank) after changing practices; thus, this was not so much about deciding whether it was worth changing the practices as it was about being satisfied by seeing the difference it made to change practices the way he did. This satisfaction by the financial calculations was as much related to environmental concerns and striving for self-sufficiency.

### 3.3. Do Households Change Engagement Related to Energy Consumption by Being Prosumers?

Finally, the questionnaire also asked households with PVs whether having a PV means that they are more environmentally conscious, more aware of saving energy, more interested in the energy system, and more attentive towards their electricity consumption. As shown in Figure 9, most households somewhat or strongly agreed with all these statements. Thus, the survey provides a clear picture that PV owners consider themselves more engaged, aware, concerned, and interested. The answers to the question regarding whether PV owners discuss energy more is slightly more ambivalent.

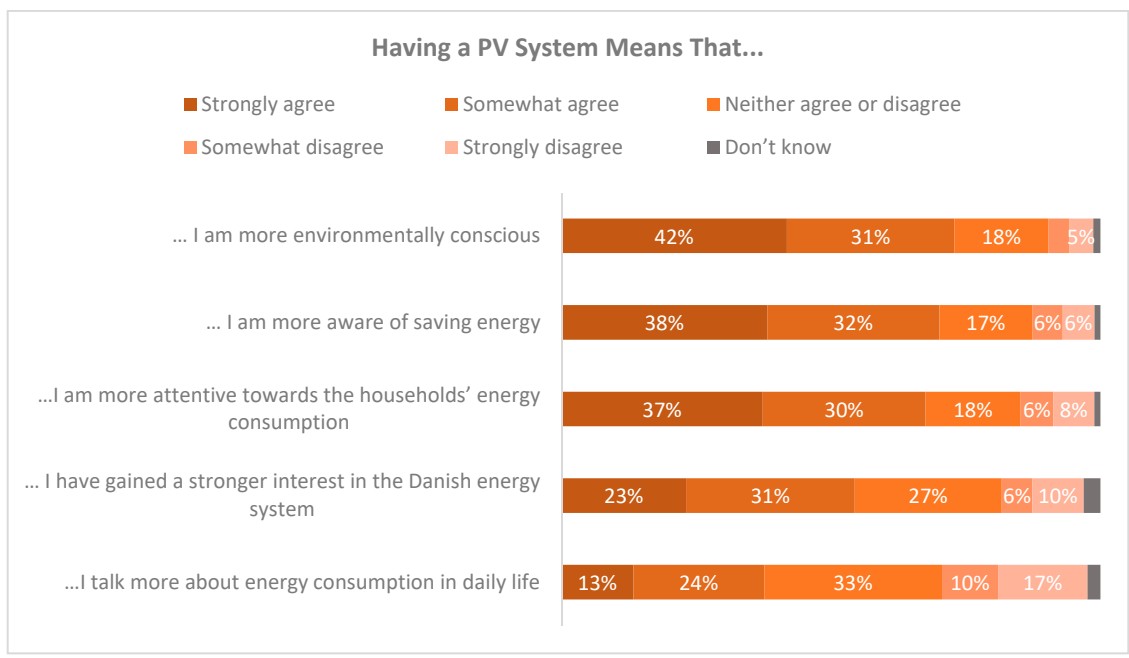

**Figure 9.** Responses to the statement 'Having a PV system means that . . . '. *n* = 2477. Percentages below 5% are not reported. The full table is in Appendix A.

As the questionnaire does not compare households before and after the acquisition of PVs and because it does not compare households using PVs with households without PVs, based on these answers, it may be difficult to determine whether households change their attitudes because of the PVs; however, the self-reported answers suggest this tendency.

The qualitative interviews indicate that all the interviewed members of the households were generally interested and knowledgeable about energy systems and resource consumption due to being a prosumer to some extent. As expressed by carl: 'I always think we've had [an awareness]. But, it has become much more ongoing now, one might say'.

As this quote from Carl suggests and as we know from the survey [27], households that buy PVs often already have an interest in energy issues before they buy PVs. However, the PV system may work by extending and maintaining this interest. Furthermore, the qualitative interviews suggested that the participants were conscious of not only the electricity system and PV production but equally of water and heat consumption. As an example, virtually all 12 interviewed households could roughly

remember their consumption and explain why it was high, low, or declining and how their water and heat consumption correlated with their electricity consumption. Further, it is relevant to notice that one of the interviewed households had the PVs because they bought a new build house where PVs were already planned. This household thus obtained PVs without deciding to buy them and without already being interested in energy issues. However, this household also discussed how PVs made them more engaged in energy issues.

## 4. Discussion

### 4.1. Amount of and Reasons for Time-Shifting and Self-Consumption

The main questions in this paper are the extent to which Danish PV prosumers time-shift their everyday practices, and if so, why they do it and what it means to them. The results reveal that most prosumers adjust their everyday practices to use their produced electricity. More specifically, this paper documents that especially dishwashing, laundry, and drying are 'always' time-shifted by 20% to 26% of PV owners, whereas an additional 38% to 50% of PV owners 'often' time-shift their use of dishwashers, washing machines, and dryers (Figure 1). The primary reasons given for time-shifting are equally divided between financial gain, environment issues, and self-sufficiency (Figure 2). Based on the qualitative interviews, informants do not separate the three reasons from each other. For instance, financial gain is also considered a marker of self-sufficiency and an environmentally sound behaviour.

Factors that promote or prevent time-shifting include that it is important that it fits into the everyday routines of the household and that someone is home during the daytime. Technological possibilities, such as timer functions, were also indicated to be relevant, though to a slightly lesser extent (Figures 2 and 3). The results further demonstrate a significant difference between different metering schemes in the extent to which households time-shift their energy-consumption practices. Households that are on immediate or hourly metering and have a financial incentive to time-shift, time-shift their consumption to a significantly larger degree compared to households on annual accounting. Among households on immediate metering, 39% stated that they adjust their electricity consumption to their PV production 'to a great extent', whereas only 7% of the households on annual metering adjust consumption 'to a great extent', and 46% on annual metering stated that they never time-shift to their production.

Thus, the metering scheme makes a difference; however, the qualitative interviews revealed that this should not be understood in a simple rational economic sense. In line with this, recent studies have considered economics in everyday life and have given explanations for how prices may influence practices in ways other than in an economically rational approach [31–34]. For instance, Strengers argued that price can convey meanings to consumers, for instance, meanings of scarcity or abundance [34]. In our case, the meaning that is conveyed by the price difference is related to questions of self-sufficiency; using one's produced electricity is given meaning through the difference in price between what prosumers can sell electricity for and what prosumers must pay for the electricity they buy. Furthermore, as described in [31,32], the price signal should be viewed in interaction with other elements of everyday practices than just meanings, such as elements of competence and materiality in the form of device interaction, with the role a given price signal has in the performance of practices. Our point is that the interviewed households did not make any calculations before time-shifting; still, the meanings related to the net-metering scheme made many participants time-shift quite often.

Comparing the results of time-shifting and self-consumption directly to previous studies is difficult because the methods and survey questions are different among different studies. A Swedish study found little time-shifting [19], although this was also related to a metering scheme with no financial incentive to time-shift, whereas a study from Wallonia found that even without an incentive, as many as 40% declared that they time-shifted [21]. A study from Germany that included households both with and without incentives, similar to our study, found this to be of importance [22], although their results are not directly comparable to ours because of differences in metering schemes and analysis methods.

Further, a study using actual time-based metered production and consumption data estimated 45% self-consumption among UK PV owners [35], which is interesting but not comparable to our study because we did not include metered production and consumption data.

## 4.2. Time-Shifting and Self-Consumption in a Research and Policy Perspective

Time-shifting is widespread among PV owners, which is challenging the perspective that energy is consumed routinely and without interest in energy. This should be included in future research on how to understand the way everyday practices change. Further, this paper reveals that households with PVs consider themselves more aware, concerned, and interested in energy consumption, which is in line with the findings of other recent studies [19,20]. However, the presented data only relate to households using PVs and do not compare a before and after situation.

It is documented that buying PVs relates strongly to being interested in energy, and data show how this interest is made stronger by owning PVs. The analysis cannot determine whether consumers with PVs but without any previous interest in energy would also be engaged in energy just by owing a PV system. However, one of the interviewees indicated that this may be the case. It is relevant from a policy perspective to understand whether PVs can mainly support those already having interest and a technical understanding of energy or whether PVs also work to engage people not previously interested or knowledgeable in energy. The register analysis of all households with PVs in Denmark indicates an overrepresentation of technically and well-educated households among PV owners [23]. Thus, it is relevant to know the extent other groups not previously engaged in energy issues can also become motivated to engage in energy by being a prosumer. In the future, when more newly built houses have PVs and when people buy older houses already equipped with PVs, empirical possibilities will exist to statistically investigate whether becoming a prosumer by chance rather than on purpose makes prosumers more interested in energy.

Time-shifting to self-consumption has been regarded as positive according to a grid perspective by some researchers [21,36,37]; however, it can also be argued that self-consumption is a sub-optimisation on a micro-level in contrast to focusing on balancing the overall system [38]. PV self-consumption lead prosumers to time-shift to the daytime when they produce the energy themselves, rather than to nighttime, which may be better for the energy system. The Danish political reasoning behind immediate or hourly consumption, given the incentive to use one's produced electricity, was to stop the overly financially attractive scheme, which the government was losing money on because of the lack of energy tax. The result was that PV owners on immediate and hourly net-metering schemes felt a strong incentive to time-shift to their production rather than shift to nighttime use, which could be more beneficial from a system perspective and a more general sustainability agenda.

To our knowledge, very little research interest exists on this subject, although it was also mentioned in another study [20]. More interest both from policy and researchers on this subject is recommended. Researchers could compare the extent to which households time-shift according to variable tariffs, such as with lower night tariffs, and the extent to which they time-shift to their production (if they are on immediate or hourly net-metering schemes) if the monetary benefits are at the same level. This could reveal whether the meanings relating time-shifting to self-sufficiency vs. time-shifting to grid optimization have different effects with the same financial gain.

## 4.3. Limitations of this Study

The results presented in this paper are strong in the sense that we have a large representative sample for statistical analysis, including households with and without an economic incentive. In addition, the inclusion of qualitative interviews to gain deeper insight into meanings is a strength of this study. The sample of qualitative interviews is biased towards older households on the annual metering scheme, thus representing households with less incentive to time-shift and those who were at home during the daytime who have a better opportunity to time-shift. Further limitations of the present study include that all statistical data are self-reported, and no metered production or consumption

data are included. In addition, the data do not include before and after buying PVs or include households without PVs. Finally, this is a case study based on only one country, and the results should not be interpreted as representative of a global perspective. Rather, it should offer inspiration for cross-national comparison in future studies.

## 5. Conclusions and Recommendations

A future low-carbon society requires engaged consumers, as research has long shown that technology alone cannot deliver the necessary solutions. Promoting prosumption is a way to engage consumers in energy issues, such as time-shifting. Thus, the actual renewable energy, which is produced by prosumers, is a benefit to the sustainable transition of the energy system, and the influence it may have on consumers' engagement and practices is important. Based on the results presented in this paper, which include both representative survey data and in-depth interview data on PV owners, the question of whether being a prosumer is a technique to engage households in more sustainable energy consumption can be answered with a clear yes.

Previous research has been ambivalent regarding the question of the extent to which prosumers time-shift their consumption. The results presented in this paper build on a strong statistically representative sample of Danish PV owners, which includes households with and without a financial incentive to time-shift their consumption. Moreover, the results support previous research, which has stated that households with a financial incentive time-shift to a large degree, especially concerning the use of dishwashers and washing machines. However, the results also confirm the findings of previous studies that indicate that households without a financial incentive also time-shift these practices, though our results show they do so to a significantly lesser degree compared to households with financial incentives. Based on the qualitative interviews, this should not be interpreted as simply an outcome of rational economic behaviour; rather, how households interpret financial incentives is integrated into the understanding of self-sufficiency and environmental concern.

From a policy perspective, it is relevant to be aware of prosumption to engage households. However, whether prosumers should be given financial incentives only to time-shift to their production or whether they should be incentivized to time-shift according to the needs from the grid perspective should be considered. As flexible metering pricing is rolled out, how this price structure works as an incentive should be considered together with the incentive for prosumers to consume when producing energy.

From an international perspective, the results presented in this paper are from a specific geographical and national context. The results could be expected to be transferable to other north European countries, although variations in the results from a global perspective should be expected, which calls for studies to include cross-cultural comparisons. It is also recommended that future research, rather than relying on self-reported data on time-shifting, should also include actual metered data on production and consumption.

**Author Contributions:** Conceptualization, K.G.-H.; Data curation, A.R.H. and M.M.; Formal analysis, K.G.-H., A.R.H. and M.M.; Funding acquisition, K.G.-H.; Investigation, K.G.-H., A.R.H. and M.M.; Methodology, K.G.-H., A.R.H. and M.M.; Project administration, A.R.H.; Validation, A.R.H.; Writing–original draft, K.G.-H.; Writing–review & editing, K.G.-H., A.R.H. and M.M. All authors have read and agreed to the published version of the manuscript.

**Funding:** This research was funded by ForskEL, grant number 2016-1-12504.

**Acknowledgments:** We would like to thank the households participating in the interviews, the two net companies we have worked together with on this project, and our research group Sustainable Cities and Everyday Practices for the valuable comments on a previous version.

**Conflicts of Interest:** The authors declare no conflict of interest.

## Appendix A

**Table A1.** Responses to the question "How often do you adjust your use of the following appliances to utilize your own produced electricity (for example using a timer)?".

|  | Always | Often | Sometimes | Rarely | Never | Don't know |
|---|---|---|---|---|---|---|
| Washing machine (N = 1673) | 26.2% (439) | 40.5% (677) | 17.9% (300) | 9.9% (165) | 5.4% (90) | 0.1% (2) |
| Tumble dryer (N = 1169) | 22.1% (258) | 49.7% (464) | 19.3% (226) | 12.2% (142) | 6.5% (76) | 0.3% (3) |
| Dishwasher (N = 1600) | 21.4% (343) | 37.6% (602) | 19.9% (318) | 14.4% (231) | 6.5% (104) | 0.1% (2) |
| Vacuum cleaner (N = 1675) | 15.4% (258) | 23.4% (392) | 29.9% (334) | 23.0% (386) | 18.1% (303) | 0.1% (2) |
| Charging of electrical appliances (e.g. mobile) (N = 1663) | 8.7% (144) | 19.4% (322) | 23.3% (388) | 27.0% (449) | 21.4% (356) | 0.2% (4) |
| Cooker/oven (N = 1670) | 7.3% (122) | 18.7% (312) | 24.4% (408) | 31.6% (527) | 17.8% (298) | 0.2% (3) |

Note: Numbers of responses in parentheses.

**Table A2.** Responses to the question "How important is the following for your household to move the time of using appliances to utilize own produced power? N = 1674.

|  | 1. Very Important | 2 | 3 | 4 | 5. Not Important At All | Don't Know |
|---|---|---|---|---|---|---|
| That it provides a financial gain | 32.4% (542) | 28.1% (470) | 24.8% (415) | 9.4% (157) | 4.0% (67) | 1.4% (23) |
| That it is good for the environment | 31.5% (528) | 32.4% (543) | 26.2% (439) | 6.1% (102) | 2.3% (39) | 1.4% (23) |
| To be self-sufficient | 30.5% (511) | 33.9% (567) | 23.7% (397) | 7.5% (126) | 2.9% (49) | 1.4% (24) |
| That it fits into the daily life of the household | 25.7% (430) | 37.9% (634) | 23.9% (400) | 5.85% (98) | 4.5% (75) | 2.2% (37) |
| 'That there is someone at home during the daytime | 24.3% (407) | 28.6% (478) | 22.3% (374) | 11.4% (191) | 10.1% (169) | 3.3% (55) |
| That there is agreement in the household to move routines | 16.3% (272) | 27.3% (457) | 28.5% (477) | 13.3% (222) | 9.8% (164) | 4.9% (82) |
| Good technical options for moving routines, such as 'timer' functions | 17.1% (287) | 22.8% (382) | 24.0% (401) | 15.2% (254) | 12.1% (203) | 8.8% (147) |

Note: Numbers of responses in parentheses.

**Table A3.** Responses to the question "How important is the following for your household *not* to move the time of using appliances to utilize own produced power? N = 2834.

|  | 1. Very Important | 2 | 3 | 4 | 5. Not Important At All | Don't Know |
|---|---|---|---|---|---|---|
| That it does not fit into the daily life of the household | 18.1% (331) | 23.8% (436) | 25.5% (450) | 10.4% (191) | 14.3% (262) | 8.9% (164) |
| That there is nobody at home during the daytime | 18.1% (332) | 19.0% (348) | 18.3% (335) | 12.5% (230) | 22.0% (403) | 10.1% (186) |
| That the economic gain is missing or too little | 15.2% (279) | 18.7% (343) | 24.8% (455) | 16.1% (296) | 14.4% (264) | 10.7% (197) |
| That it is not technically possible to change everyday routines (e.g., if appliances do not have timer function) | 10.3% (188) | 14.2% (261) | 24.6% (451) | 15.5% (284) | 21.4% (393) | 14.0% (257) |
| That I do not see a reason to do it | 8.2% (150) | 11.2% (206) | 23.0% (421) | 18.7% (348) | 24.1% (442) | 14.6% (267) |
| That it is too cumbersome to change every day routines | 7.2% (132) | 14.1% (259) | 27.3% (501) | 20.9% (384) | 20.6% (377) | 9.9% (181) |

Note: Numbers of responses in parentheses.

**Table A4.** Responses to the question regarding when households report using a dishwasher across the net-metering scheme.

| Dishwasher | Day (6 a.m. to 5 p.m.) | Evening (5 p.m. to 11 p.m.) | Night (11 p.m. to 6 a.m.) |
|---|---|---|---|
| Hourly (N = 470) | 67.5% (317) | 36.2% (170) | 6.8% (32) |
| Immediately (N = 732) | 65.2% (477) | 36.6% (268) | 3.55% (26) |
| Annually (N = 1275) | 35.5% (453) | 65.0% (829) | 12.3% (157) |

**Table A5.** Responses to the question regarding when households report using a washing machine across the net-metering scheme.

| Washing Machine | Day (6 a.m. to 5 p.m.) | Evening (5 p.m. to 11 p.m.) | Night (11 p.m. to 6 a.m.) |
|---|---|---|---|
| Hourly (N = 470) | 86.4% (406) | 22.6% (106) | 4.9% (23) |
| Immediately (N = 731) | 88.3% (646) | 18.3% (134) | 1.8% (13) |
| Annually (N = 1275) | 75.0% (956) | 36.1% (460) | 6.0% (76) |

**Table A6.** Answers to the question "To what extent do your household adjust your electricity consumption to own PV production" (N = 2477).

| Net Metering Scheme | To a Great Extent | To Some Extent | To a Small Extent | Not At All | Don't Know |
|---|---|---|---|---|---|
| Hourly | 36.6% (172) | 38.3% (180) | 12.6% (59) | 10.2% (48) | 2.3% (11) |
| Immediately | 38.9% (285) | 37.0% (271) | 12.0% (88) | 9.8% (72) | 2.2% (16) |
| Annually | 6.6% (84) | 19.1% (244) | 22.9% (292) | 45.6% (581) | 5.8% (74) |

**Table A7.** Answers to the question "To what extent do your household adjust your electricity consumption to peak demands in electricity grid" (N = 2477).

| | To a Great Extent | To Some Extent | To a Small Extent | Not At All | Don't Know |
|---|---|---|---|---|---|
| Hourly | 1.5% (7) | 15.3% (72) | 37.9% (178) | 49.8% (187) | 5.5% (26) |
| Immediately | 2.6% (19) | 13.7% (100) | 34.2% (250) | 43.0% (315) | 6.6% (48) |
| Annually | 1.6% (20) | 15.1% (192) | 28.2% (359) | 49.4% (630) | 5.8% (74) |

**Table A8.** Responses to the questions "Having a PV system means that ...".

| | Strongly Agree | Somewhat Agree | Neither Agree or Disagree | Somewhat Disagree | Strongly Disagree | Don't Know |
|---|---|---|---|---|---|---|
| ... I am more environmentally conscious | 41.8% (1035) | 31.1% (771) | 17.5% (433) | 3.8% (94) | 4.5% (111) | 1.3% (33) |
| ... I am more aware of saving energy | 38.2% (945) | 32.0% (793) | 17.3% (429) | 5.6% (138) | 5.9% (146) | 1.1% (26) |
| ... I am more attentive towards the households' energy consumption | 37.3% (923) | 30.2% (749) | 17.6% (436) | 6.1% (152) | 7.6% (189) | 1.1% (28) |
| ... I have gained a stronger interest in the Danish energy system | 23.1% (573) | 31.3% (775) | 27.3% (677) | 5.7% (142) | 9.5% (234) | 3.1% (76) |
| ... I talk more about energy consumption in daily life | 13.3% (331) | 24.3% (601) | 33.0% (818) | 10.4% (257) | 16.6% (410) | 2.4% (60) |

Note: Numbers of responses in parentheses. N = 2477.

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
