# Peer review of "Danish PV Prosumers’ Time-Shifting of Energy-Consuming Everyday Practices"

_sustainability, doi:10.3390/su12104121_

Round 1

Reviewer 1 Report

The authors conducted an interesting study on prosumers energy-use behaviours at houses in Denmark.

My comments are as follows:

Abstract: Please add a few numerical findings in the abstract.

Keywords: ‘demand respond management’ is not a correct keyword.

The present literature review dominantly focused on the energy use behaviour of the prosumers in the developed world. What about developing countries? For instance, on page 2, line 58-60: “Furthermore, this study found that some households started to use more of their “free” electricity rather than giving is (it?) away, as they expressed it.”- In contrast, a different finding is reported for a developing country’s prosumers. Actually, it depends on the context. This should also be highlighted. For further detail please check- https://doi.org/10.1016/j.erss.2019.04.019

Overall, the literature review has failed to establish the research gap that there is a true need for this study. For instance, the link between demand-side management (DSM) and prosumers is missing. It is recommended to conduct a thorough literature review considering DSM and human behaviour towards in-house energy use. In addition, please consider prosumers energy practices for both developed and developing countries. Few examples for your reference (please search for more):

https://doi.org/10.3390/app6100275

https://doi.org/10.3390/en10111771

https://doi.org/10.1016/j.energy.2017.10.068

https://doi.org/10.1016/j.scs.2017.09.009

https://doi.org/10.1016/j.rser.2017.07.018

https://doi.org/10.1016/j.erss.2018.04.006

For energy-saving behaviour and DSM:

https://doi.org/10.1016/j.erss.2014.04.008

https://doi.org/10.1007/s40095-019-0302-3

https://doi.org/10.1016/j.egypro.2013.11.015

https://doi.org/10.1016/j.egypro.2015.06.032

Materials and Methods: I would suggest adding the used questionnaire in the appendix/as supplementary materials. A flowchart showing each step of the methodology used would be a good addition to understand the whole process in brief.

What were the capacities of the surveyed households’ solar panels (2,505)? A box and whisker plot is highly recommended for this. This will show the maximum and minimum capacity along with average and median values.

A list of common home appliances for the surveyed households should also be added.

For the interviews, a table showing their demographics would be helpful.

Line: 141- “…12 out of the 18 were pensioners…”.  About 67% were pensioners. Thus, the obtained result would be definitely biased. How did you deal with this? A clear explanation needs to be included regarding this. Do you think, it’s a limitation of this study? If so, how future research could overcome this limitation?

Results: Have you found any rebound effect (increased energy consumption) in households that participated in prosumerism? If such data were available.

A separate discussion section is recommended to compare the results of this study and other studies in the literature, which is partially done at present. Importantly, please check- is your finding inline with the other findings in the literature who were not prosumers but adopted the energy-saving behaviour (such as time-shifting of the washing machine) a part of DSM at home.  

Figure 2: It is difficult to distinguish the colors. Please use different textures for the figure.

What were the limitations of this study? What are the future research scopes that can be drawn from this study?

Author Response

Review comments

Answer

Abstract: Please add a few numerical findings in the abstract

Have been added

Keywords: ‘demand respond management’ is not a correct keyword.

Thanks for noticing – have been changed

The present literature review dominantly focused on the energy use behaviour of the prosumers in the developed world. What about developing countries? For instance, on page 2, line 58-60: “Furthermore, this study found that some households started to use more of their “free” electricity rather than giving is (it?) away, as they expressed it.”- In contrast, a different finding is reported for a developing country’s prosumers. Actually, it depends on the context. This should also be highlighted. For further detail please check- https://doi.org/10.1016/j.erss.2019.04.019

We have added this reference and have argued that our perspective is the EU north area, and that there may be important socio-demographic difference to other parts of the world.

Overall, the literature review has failed to establish the research gap that there is a true need for this study. For instance, the link between demand-side management (DSM) and prosumers is missing. It is recommended to conduct a thorough literature review considering DSM and human behaviour towards in-house energy use. In addition, please consider prosumers energy practices for both developed and developing countries. Few examples for your reference (please search for more):

https://doi.org/10.3390/app6100275

https://doi.org/10.3390/en10111771

https://doi.org/10.1016/j.energy.2017.10.068

https://doi.org/10.1016/j.scs.2017.09.009

https://doi.org/10.1016/j.rser.2017.07.018

https://doi.org/10.1016/j.erss.2018.04.006

For energy-saving behaviour and DSM:

https://doi.org/10.1016/j.erss.2014.04.008

https://doi.org/10.1007/s40095-019-0302-3

https://doi.org/10.1016/j.egypro.2013.11.015

https://doi.org/10.1016/j.egypro.2015.06.032

Thanks for the links, many of these and others have been added.

Materials and Methods: I would suggest adding the used questionnaire in the appendix/as supplementary materials. A flowchart showing each step of the methodology used would be a good addition to understand the whole process in brief.

Additional material have been added as appendix

What were the capacities of the surveyed households’ solar panels (2,505)? A box and whisker plot is highly recommended for this. This will show the maximum and minimum capacity along with average and median values.

Have been added, just as one average number and the standard deviation in Materials and Methods section

 list of common home appliances for the surveyed households should also be added.

We are not really sure what the reviewer have in mind here, and think it would be too much, and a bit out of scoop with information on what appliances all households have. Other reviewers have argued the paper I s already too long

For the interviews, a table showing their demographics would be helpful.

Has been added

Line: 141- “…12 out of the 18 were pensioners…”.  About 67% were pensioners. Thus, the obtained result would be definitely biased. How did you deal with this? A clear explanation needs to be included regarding this. Do you think, it’s a limitation of this study? If so, how future research could overcome this limitation?

Comments and discussion have been added on this in methods and discussion

Results: Have you found any rebound effect (increased energy consumption) in households that participated in prosumerism? If such data were available.

As we did not include energy data in this paper, we are not able to document this sufficiently

A separate discussion section is recommended to compare the results of this study and other studies in the literature, which is partially done at present. Importantly, please check- is your finding inline with the other findings in the literature who were not prosumers but adopted the energy-saving behaviour (such as time-shifting of the washing machine) a part of DSM at home.  

We have reworked the paper and now include both a discussion and a conclusion

Figure 2: It is difficult to distinguish the colors. Please use different textures for the figure.

All figures have been reworked

What were the limitations of this study? What are the future research scopes that can be drawn from this study?

A section on this have been added

Reviewer 2 Report

In some parts of the Manuscript modification/fine tuning would be beneficial to provide more clarity to the readers:

line 16 <... more aware of energy ...> - what do authors have in mind - energy as a concept or how improve energy efficiency/reduce consumption/etc.?

line 18 <Results discuss how this ...> - can results discuss in practise?

line 30-33 <The EU SET (Strategic Energy Technology) Plan for instance ...> clumsy sentence.

line 33 <This paper is interested ...> - can paper (written document) be interested in? 

line 34 <... households’ understanding of energy,> - what authors have in mind?

line 52-53 <... that prosumers become interested in and engaged in energy issues, even many of them 52 already before acquiring the PVs also was engaged.. - unclear sentence.

line 70-72 <This has a major impact on how to understand how smart grid technologies and flexible consumption are integrated into the everyday life of households, and how to understand the importance of having PVs for the household's relation to energy consumption.> - seems that the sentence contradicts the previous sentence on ordinary consumer attitude and understanding in energy production. 

line 76 <... under different times, ....> - it could be understood two-fold.

In the entire text should be cheched words "prize", "seperated", "satiesfyed", "seening", "thir" for spelling.

line 87 <... resulted in a massive loss for the state> - massive loss of what?

line 96 <... relevant to include ...> - unclear

line 101-102 <Thus in an international perspective results from Denmark should be interpreted in the light of this.> - unclear sentence.

line 103 <... reviewed knowledge the main aim of this paper is to investigate ...> - unclear. Does the paper investigate something? In the paper investigation results are presented.

line 105-106 <The paper will first report on the methods used in the project, and following this is results and discussion.> - unclear sentence

In the Introduction part authors could have probably taken a look also in newer studies and papers. Do exist only Swedish and Danish studies on economic incentives and engagement of prosumers?

line 112-114 <The presented material including its results is described in more details in two Danish reports on respectively the survey questionnaire 19 and the qualitative material 20. Further, another paper is published on the survey results, highlighting different ways of being prosumer 21.> - seems that quite a bit already is published. What is new in presented manuscript?

line 125 - what Stata stands for?

line 143 <... and one were one hourly ...> - spelling

line 151-153 <Results presented in this paper focus on the question of the extent to which being prosumer makes a difference for the engagement in energy issues and for time shifting of everyday practices. The following will present results divided into three subsections answering question of respectively> - unclear sentences

line 272, 293, 347, 376 <is seen> would be better rephrase - <presented>

line 290 <This figure compare the question ... > - figure does not compare. At the best it can present.

line 301-302 <This is actually surprising, given that half of the respondents stated that they “have gained a stronger interest in the Danish energy system” (see Table 1). > - how this statement correlates to Table 1?

line 332 - (APP WATSS) stands for?

line 349 - Could the figure 2 be more colourful?

line 404-406 <A future low-carbon society do need engaged consumers, as research for long have showed that technological solutions cannot stand alone or be viewed isolated from the practices and engagements of households.>  -unclear sentence

Since the Danish situation is presented in the manuscript, would authors consider somehow mention that in the title?

Author Response

Reviewer 2

Reviewer comments

answers

In some parts of the Manuscript modification/fine tuning would be beneficial to provide more clarity to the readers:

line 16 <... more aware of energy ...> - what do authors have in mind - energy as a concept or how improve energy efficiency/reduce consumption/etc.?

line 18 <Results discuss how this ...> - can results discuss in practise?

line 30-33 <The EU SET (Strategic Energy Technology) Plan for instance ...> clumsy sentence.

line 33 <This paper is interested ...> - can paper (written document) be interested in? 

line 34 <... households’ understanding of energy,> - what authors have in mind?

line 52-53 <... that prosumers become interested in and engaged in energy issues, even many of them 52 already before acquiring the PVs also was engaged.. - unclear sentence.

line 70-72 <This has a major impact on how to understand how smart grid technologies and flexible consumption are integrated into the everyday life of households, and how to understand the importance of having PVs for the household's relation to energy consumption.> - seems that the sentence contradicts the previous sentence on ordinary consumer attitude and understanding in energy production. 

line 76 <... under different times, ....> - it could be understood two-fold.

In the entire text should be cheched words "prize", "seperated", "satiesfyed", "seening", "thir" for spelling.

line 87 <... resulted in a massive loss for the state> - massive loss of what?

line 96 <... relevant to include ...> - unclear

line 101-102 <Thus in an international perspective results from Denmark should be interpreted in the light of this.> - unclear sentence.

line 103 <... reviewed knowledge the main aim of this paper is to investigate ...> - unclear. Does the paper investigate something? In the paper investigation results are presented.

line 105-106 <The paper will first report on the methods used in the project, and following this is results and discussion.> - unclear sentence

The paper have been rewritten according to language issues and clarity of wording, and have been copy edited by a native speaker. All of these sentences have thus been changed

In the Introduction part authors could have probably taken a look also in newer studies and papers. Do exist only Swedish and Danish studies on economic incentives and engagement of prosumers?

More references have been added

line 112-114 <The presented material including its results is described in more details in two Danish reports on respectively the survey questionnaire 19 and the qualitative material 20. Further, another paper is published on the survey results, highlighting different ways of being prosumer 21.> - seems that quite a bit already is published. What is new in presented manuscript?

It is stated more clearly now what the new in this paper is, which is both the analysis on survey data and the combination of qualitative and quantitative material

line 125 - what Stata stands for?

Now rewritten

line 143 <... and one were one hourly ...> - spelling

The paper have been rewritten according to language issues and clarity of wording, and have been copy edited by a native speaker.

line 151-153 <Results presented in this paper focus on the question of the extent to which being prosumer makes a difference for the engagement in energy issues and for time shifting of everyday practices. The following will present results divided into three subsections answering question of respectively> - unclear sentences

The paper have been rewritten according to language issues and clarity of wording, and have been copy edited by a native speaker

line 272, 293, 347, 376 <is seen> would be better rephrase - <presented>

Thanks, have been changed

line 290 <This figure compare the question ... > - figure does not compare. At the best it can present.

The paper have been rewritten according to language issues and clarity of wording, and have been copy edited by a native speaker

line 301-302 <This is actually surprising, given that half of the respondents stated that they “have gained a stronger interest in the Danish energy system” (see Table 1). > - how this statement correlates to Table 1?

Sorry, this is a wrong table number, have been changed

line 332 - (APP WATSS) stands for?

Have been changed

line 349 - Could the figure 2 be more colourful?

All figures have been reworked

line 404-406 <A future low-carbon society do need engaged consumers, as research for long have showed that technological solutions cannot stand alone or be viewed isolated from the practices and engagements of households.>  -unclear sentence

Have been rewriten

Since the Danish situation is presented in the manuscript, would authors consider somehow mention that in the title?

Have been added

Reviewer 3 Report

This paper : “PV prosumers’ time-shifting of energy-consuming everyday practices” deals with the important question of consumers' involvement in the development of renewable energy, particularly photovoltaic solar energy. The concept of the paper is really interesting and can be a great tool for policy makers, for instance.

Nevertheless, this paper suffers of important flaws. The English language should be profoundly revised. The paper is poorly organized, there is no balance between the different parts and it is too long. For example one of the most important aspects of renewable energy, the motivation of consumers: generally environmental or purely economic, based on incentives, is only given a little attention at the end of the paper. The results are presented without any analysis.

Despite the importance of the subject, we do not recommend this paper to be accepted in this form. It needs profound revision before being considered for publication.

Author Response

Reviewer comments

answers

This paper : “PV prosumers’ time-shifting of energy-consuming everyday practices” deals with the important question of consumers' involvement in the development of renewable energy, particularly photovoltaic solar energy. The concept of the paper is really interesting and can be a great tool for policy makers, for instance.

Thanks!

Nevertheless, this paper suffers of important flaws. The English language should be profoundly revised.

The paper have now been copy edited by a native speaker

The paper is poorly organized, there is no balance between the different parts and it is too long. For example one of the most important aspects of renewable energy, the motivation of consumers: generally environmental or purely economic, based on incentives, is only given a little attention at the end of the paper.

Large parts of the paper have been rewritten, so hopefully it now stands clearer. The paper is, however, not shortened, as no other reviewers suggested this, and as other reviewers actually suggested adding stuff.

Questions related to the motivation of consumers: generally environmental or purely economic, is now written more clearly into introduction and research questions, and then linked to the discussion and conclusion

The results are presented without any analysis

The results section include descriptive statistics from survey combined with qualitative analysis of interviews. In the research tradition we come from, this means that the results are presented with analysis.

Reviewer 4 Report

The article is interesting and well written. It also shows important findings in the case study, both qualitatively and quantitatively.

However, the presentation of quantitative results should be improved. Therefore I will suggest the following:

  • The tables contained too much information. Visualizing the tables, e.g., with pie chart diagram, is suggested for better readability. The tables should be put in the appendix section.
  • The existing and to-be-added figures should be colored, instead of in greyscale. 

Author Response

Review 4

Reviewer comments

answers

The article is interesting and well written. It also shows important findings in the case study, both qualitatively and quantitatively

Thanks!

The tables contained too much information. Visualizing the tables, e.g., with pie chart diagram, is suggested for better readability. The tables should be put in the appendix section.

Thanks for the advice, this has now been changed

The existing and to-be-added figures should be colored, instead of in greyscale. 

Have been changed

Reviewer 5 Report

This manuscript gives a significant example dealing with the issue of being a committed player in the daily management of one's energy consumption. The authors rely on extensive data collected from a representative sample of 2,505 PV system owners in Denmark. About ten in-depth qualitative interviews were also conducted to complete this study. The proposed methodology is clearly explained. The numerous results show that Danish PV system owners themselves consider that they have become more energy conscious by adapting their energy consumption.

The written expression is fluid, which makes the document very pleasant to read.

I propose the following minor changes to improve the quality of the document:

  • Introduction:
    • The authors focused their study on the owners of photovoltaic power plants. Why this choice? Please situate your case in relation to the growth of photovoltaics in the European Union, Asia, North America and the rest of the world. What about owners using mixed solar PV and wind power solutions?
    • Your literature review is consistent and up to date. On the other hand, these types of issues have probably been addressed in MDPI journals. Please insert some of them in the references already cited. Please also use only English language references.
    • Please make your contributions more visible before the manuscript outline appears.
    • The document plan needs to be rewritten.
  • Materials and Methods: A general outline is needed to illustrate the methodology described in this article.
  • Results and Discussion:
    • The quality of Figure 1 is insufficient. Please rework this figure.
    • I think we need to distinguish the discussion of the results from the general conclusion of the manuscript. A summary table of key determining outcomes would be welcome in the discussion. Please position your results in relation to the results of studies carried out e.g. within the framework of European projects.
    • Your general conclusion should recall the problematic of the document, synthesize the main ins and outs and give some research perspectives.

Author Response

Reviewer comments

Answers

The written expression is fluid, which makes the document very pleasant to read.

Thanks!

Introduction: The authors focused their study on the owners of photovoltaic power plants. Why this choice? Please situate your case in relation to the growth of photovoltaics in the European Union, Asia, North America and the rest of the world. What about owners using mixed solar PV and wind power solutions?

We have now situated our case better in the introduction

Introduction: Your literature review is consistent and up to date. On the other hand, these types of issues have probably been addressed in MDPI journals. Please insert some of them in the references already cited. Please also use only English language references

We have added references to the introduction. We still include a few Danish references, which is the reports, which presents other parts of our study. These are only in Danish, and we need to make reference to these

Introduction: Please make your contributions more visible before the manuscript outline appears.

We have reframed introduction to more clearly show our contribution to the existing literature

Introduction: The document plan needs to be rewritten.

We are sorry to admit that we are not sure what the reviewer mean – maybe because we come from different research traditions.  We have reframed the research question of the paper and we have described the data and methods used, in a way which is common in our research tradition.  

Materials and Methods: A general outline is needed to illustrate the methodology described in this article.

We are again sorry that we are not sure what the reviewer mean.  We have reframed parts of the methods section and added supplementary material, which may answer to this

Results and Discussion: The quality of Figure 1 is insufficient. Please rework this figure.

All figures have been reworked

Results and Discussion: I think we need to distinguish the discussion of the results from the general conclusion of the manuscript. A summary table of key determining outcomes would be welcome in the discussion. Please position your results in relation to the results of studies carried out e.g. within the framework of European projects.

We have reworked the paper and now include both a discussion and a conclusion. Further the results are also positioned in relation to other studies

Results and Discussion: Your general conclusion should recall the problematic of the document, synthesize the main ins and outs and give some research perspectives.

Conclusion have been reworked

Round 2

Reviewer 1 Report

Minor corrections are required:

Figures: please reduce the font size used in the figures, mainly the heading/title of the figures. For example, check Figure 8.

Figure 2 & 3: What about the legends 2,3, and 4? I did not find any text for these in the figure. A figure should convey a complete picture.

Author Response

Dear reviewer

Thanks for reviewing our paper again.

We have now, as advised reduced font size in the heading of figures.

As regard the legend in Figure 2 and 3, the problem of changing the legend is that the questionnaire asked to indicate a number between 1 an 5 and only the numbers 1 and 5 were having a text, so we cannot in the legend write differently. We have now in the figure text to Figure 2 and 3 written "Questionnaire asked to indicate a number from 1 (very important) to 5 (not important at all).

With these changes we think we have further improved the paper, and we hope you will now find it publishable

kind regards Kirsten Gram-Hanssen

Reviewer 3 Report

This manuscript has been substantially modified based on previous recommendations, but it could be improved to reach a much higher standard. It is nevertheless in an acceptable form to be released for publication. It brings up an interesting point regarding the development of renewable energy, it is of high interest for potential readers, particularly policy makers. 

Author Response

Dear reviewer

We thank you for your comments that the paper has been substantially modified based on previous recommendations, and that it is in an acceptable form to be released for publication. Further we appreciate your comments that it brings up an interesting point regarding the development of renewable energy, and is of high interest for potential readers, particularly policy makers.

kind regards Kirsten Gram-Hanssen